# Risk of Low Energy Availability in National and International Level Paralympic Athletes: An Exploratory Investigation

**DOI:** 10.3390/nu13030979

**Published:** 2021-03-18

**Authors:** Kelly Pritchett, Alicia DiFolco, Savannah Glasgow, Robert Pritchett, Katy Williams, Trent Stellingwerff, Patricia Roney, Susannah Scaroni, Elizabeth Broad

**Affiliations:** 1Department of Health Sciences, Central Washington University, Ellensburg, WA 98926, USA; Alicia.Difolco@cwu.edu (A.D.); savannah.glasgow@gmail.com (S.G.); robert.pritchett@cwu.edu (R.P.); katy.williams@cwu.edu (K.W.); 2Canadian Sport Institute-Pacific, Victoria, BC V9E 2C5, Canada; tstellingwerff@csipacific.ca; 3Exercise Science, Physical & Health Education, University of Victoria, Victoria, BC V8P 5C2, Canada; 4Athletics Canada, Ottawa, ON K1G 6C9, Canada; patricia.roney@athletics.ca; 5Division of Nutrition Sciences, University of Illinois Urbana-Champaign, Champaign, IL 61801, USA; sscaroni12@gmail.com; 6United States Olympic and Paralympic Committee, Chula Vista, CA 91915, USA; lizbroadnutritiion@gmail.com

**Keywords:** RED-S, energy availability, Paralympic, spinal cord injury, bone health, reproductive hormones

## Abstract

(1) Background: The purpose of this study was to examine the symptoms of low energy availability (LEA) and risk of relative energy deficiency in sport (RED-S) symptoms in para-athletes using a multi-parameter approach. (2) Methods: National level para-athletes (*n* = 9 males, *n* = 9 females) completed 7-day food and activity logs to quantify energy availability (EA), the LEA in Females Questionnaire (LEAF-Q), dual energy X-ray absorptiometry (DXA) scans to assess bone mineral density (BMD), and hormonal blood spot testing. (3) Results: Based on EA calculations, no athlete was at risk for LEA (females < 30 kcal·kg^−1^ FFM·day^−1^; and males < 25 kcal·kg^−1^ FFM·day^−1^; thresholds for able-bodied (AB) subjects). Overall, 78% of females were “at risk” for LEA using the LEAF-Q, and 67% reported birth control use, with three of these participants reporting menstrual dysfunction. BMD was clinically low in the hip (<−2 z-score) for 56% of female and 25% of male athletes (4) Conclusions: Based on calculated EA, the risk for RED-S appears to be low, but hormonal outcomes suggest that RED-S risk is high in this para-athlete population. This considerable discrepancy in various EA and RED-S assessment tools suggests the need for further investigation to determine the true prevalence of RED-S in para-athlete populations.

## 1. Introduction

Low energy availability (LEA) was initially described in female able-bodied (AB) athletes as the underpinning etiology of the female athlete triad (triad). This concept has recently been expanded upon and defined as relative energy deficiency in sport (RED-S), which includes a broader spectrum of health and performance outcomes in both females and males [1,2]. RED-S addresses aspects of decreased athletic performance, increased risk for injury, and serious short- and long-term negative health consequences. Health impairments such as depressed levels of estrogen, testosterone (TES), and insulin-like growth factor-1 (IGF-1) associated with LEA [3,4,5,6,7] have subsequently been linked to impaired bone health in AB athletes [6,7]. However, endocrine and bone health may be impacted by a spinal cord injury in a clinical/non-exercising population, therefore potentially making the interpretation of RED-S symptoms difficult in a para-athlete population.

LEA represents the amount of energy available for optimal health and physiological function (e.g., regular menses, endocrine production, bone remodeling, muscle protein synthesis) after energy expended from exercise (EEE) is subtracted from energy intake (EI), and normalized for fat free mass (FFM) [8]. Well-controlled laboratory-based studies in active AB females have demonstrated a minimum EA of 45 kcal·kg FFM^−1^·day^−1^ is required for optimal health, with LEA defined as <30 kcal·kg FFM^−1^·day^−1^ [3,7]. Thus, these values should be applied with caution, as there is significant measurement variability and discrepancies when assessing an individual’s EA [9]. A LEA threshold for males is yet to be established, although some emerging research suggests the threshold may be lower in males, and closer to 20–25 kcal·kg FFM^−1^·day^−1^ [5,10].

Para-athletes display variability in body composition, mobility, bone health, metabolic and neurological function, all of which can significantly impact the athlete’s energy requirements [11,12]. The heterogenous nature of this population makes it difficult to determine a generalized LEA threshold, but these differences (between AB and para-athletes, and within the para-athlete population due to injury-related factors) underscore the importance of this research in order to provide safe and effective nutrition recommendations unique to the para-athlete [11,12,13]. Egger et al. (2020) examined the prevalence of LEA in (*n* = 14) elite wheelchair athletes over a seven-day period and found a higher prevalence of LEA in female athletes than males, with LEA occurring on 73% and 30% of days, respectively [14]. However, the impact and prevalence of LEA symptoms in para-athletes needs further investigation, with the consideration of baseline effects of the athlete’s underlying impairment and the impact this may have on the assessment process [2,11]. As the Paralympic movement continues to grow, this research is warranted to provide assessment and treatment recommendations for the sports medicine team and coaches [11,14]. Therefore, this exploratory investigation aimed to examine elements of RED-S symptom prevalence and the assessment process of LEA including menstrual health, hormonal disturbances, bone mineral density (BMD), and metabolic and physiological functioning amongst national and international level para-athletes.

## 2. Materials and Methods

### 2.1. Study Design and Participants

An exploratory, descriptive study design was implemented within the daily training environment at the University of Illinois Urbana-Champaign (Urbana, IL, USA; January, 2019) (*n* = 13) and at a training camp in Daytona Beach, FL (March, 2019) (*n* = 5), where EI and EEE were assessed over 7 days as well as various direct (blood measures, dual energy X-ray absorptiometry (DXA; General Electric, Lunar iDXA, Madison, WI, USA and Hologic QDR 4500A, Bedford, MA, USA) scan) and indirect (questionnaires) RED-S indicators collected.

Male and female participants (international and national level para-athletes; ≥18 years old) were recruited from the United States and Canadian Paralympic programs, as well as the wheelchair basketball and track teams at the University of Illinois. Inclusion criteria were: presence of a physical disability. Globally, para-athletes are defined by the International Paralympic Committee (IPC) as an athlete with a visual, physical, or intellectual impairment. Exclusion criteria included subjects who were currently pregnant, experiencing menopause or were post-menopausal, and/or had current injuries preventing them from engaging in their normal training. Participants were informed about the study design before signing an informed consent. Ethical approval for this study was granted by the Human Subjects Review Committee (H18020).

### 2.2. EI and EEE Estimates

Table 1 highlights the procedures, adapted from Heikura et al. (2018), for estimating EI and EEE. Dietary intake and activity were recorded by participants, whom were instructed and reminded (via a training video) to maintain typical habits, over seven consecutive days [5,14]. Upon completion of the food log, a registered dietitian nutritionist (RDN) reviewed the food journals and had an opportunity to clarify any questions pertaining to food portions/intake from participants, and then entered data into nutrient analysis software (Elizabeth Stewart Hands and Associates (ESHA) Food Processor, Salem, OR, USA) to analyze caloric intake and macronutrients, including fiber. EEE was assessed using an activity diary [15] undertaken simultaneously with the food diary and was analyzed in conjunction with EI to assess EA. Fat free mass (FFM) was obtained via DXA scan.

### 2.3. Questionnaires: Eating Attitudes and Behaviors and Menstrual Status

Female participants completed the 25-item Low Energy Availability in Females Questionnaire (LEAF-Q), validated for use in AB athletes and previously described by Melin et al. (2014) [17]. Participants who score ≥8 are considered at risk for LEA, while participants scoring <8 are considered low risk. The 28-item Eating Disorder Examination-Questionnaire (EDE-Q; version 6.0) was used to assess the behaviors and attitudes related to disordered eating and eating disorders over the last 28 days [18], and was validated in AB subjects [19,20]. Female participants with a global score of ≥4 were classified as “at risk” and those with scores of <4 classified as “not at risk” for disordered eating behaviors. Recent evidence suggests the global score threshold for identifying risk of disordered eating behaviors is lower for males than it is for females [21,22]; therefore, a mean global score of ≥1.68 indicated the male participant was “at risk” for disordered eating behaviors [23]. Participants were asked to identify menstrual patterns, age of menarche, current or past menstrual irregularities, and number of menstrual cycles during the year as well as forms of birth control which may influence menses. Phase of the menstrual cycle was noted but not controlled for in this study.

### 2.4. Bone Mineral Density and Anthropometrics

DXA (General Electric, Lunar iDXA in Daytona FL, and Hologic QDR 4500A in Urbana) was used in all testing locations to assess fat-free, fat and bone mass, and BMD by doing an AP lumbar spine and total hip scan. The generated *Z*-scores were calculated based on AB populations, as there are currently no reference data for spinal cord injury (SCI) populations [24]. However, DXA has been suggested to provide precise, reproducible measures of BMD [25], and an appropriate method for assessing body composition for athletes with SCI [26]. To minimize the impact of hydration and glycogen variability, the DXA was performed in the morning following a regular training day with participants in a fasted and resting state.

### 2.5. Hormones

Using blood spot methodology, blood samples were obtained from all participants using a finger stick to examine whole blood for testosterone (TES), insulin-like growth factor-1 (IGF-1), progesterone, free triiodothyronine (fT_3_), and estradiol. Blood spot cards were allowed to dry for a minimum of 30 min and then were sent to be assayed at ZRT Laboratories (Beaverton, OR). This method has been shown to provide valid and reliable data with the following correlation values against serum samples: TES (R = 0.99), IGF-1 (R = 0.88), fT_3_ (R = 0.82), estradiol (R = 0.86), progesterone (R = 0.99) [27]. As Heikura et al. (2018) demonstrated significantly more career stress fractures in male athletes, low TES was defined as below 16.5 nmol.L^−1^ in the current study [5].

### 2.6. Statistical Analysis

Data were reported as mean ± standard deviation (SD) for dietary intake, blood measures, BMD, and calculated EA and all were reported descriptively. BMD was presented using *Z*-scores, where a *Z*-score < −1 indicated “increased risk for fracture” [28] and a *Z*-score < −2 indicated “clinically low” [24,29]. Frequencies were used to describe percentage of athlete’s “at risk” for LEA using the LEAF -Q and LEA calculations. Statistical significance.

## 3. Results

Eighteen para-athletes (females: *n* = 7 wheelchair track/marathon racing, and *n* = 2 basketball; males: *n* = 9 wheelchair track/marathon racing) completed the study (Table 2). No significant correlations were found for any of the dependent variables.

### 3.1. Energy Availability

No female participants were found to have low EA according to AB EA cutoff values (<30 kcal·kg^−1^ FFM·day^−1^), while three participants were considered to have moderate EA (30–45 kcal·kg^−1^ FFM·day^−1^). However, large daily fluctuations in EA existed (21.6% CV), with some participants displaying LEA within the 7 days (Figure 1).

Due to incomplete EEE data, EA could only be calculated for four of the male athletes. None of the male participants had an average weekly EA below the suggested low EA threshold of ≤25 kcal·kg^−1^ FFM·day^−1^ with a 29.3% CV for daily fluctuations [10].

### 3.2. Qualitative Questionnaires

LEAF-Q scores suggested that 78% of female participants were “at risk”, while the average score also represented an “at risk” score (8.8 ± 4.2) for LEA based on menstrual history and physiological symptoms of insufficient energy intake. The EDE-Q suggested that one female participant was “at risk” for disordered eating behavior. That participant was also considered “at risk” according the LEAF-Q score and had moderate calculated EA. In addition, 78% of female participants reported “restricting calorie intake” due to discomfort before activity (44%) or concerns about fitting into racing chair or prosthesis (33%).

The mean EDE-Q global score for the males indicated no participants were “at risk” for an eating disorder. One participant reported using exercise in a “driven” or “compulsive” way to expend calories and manipulate body composition; they reported engaging in this behavior during 3 days out of the prior 28 days. Four athletes (50%) reported multiple occasions during the prior 28 days where they ate an “unusually large amount of food”, and one participant reported 4 discrete days where binge eating occurred. One male athlete reported “restricting calorie intake” due to concerns about fitting into a racing chair.

### 3.3. BMD and Reproductive/Metabolic Function

BMD and reproductive and metabolic hormone levels are summarized for all participants in Table 3. Six female participants (67%) reported birth control use, with three of these participants reporting menstrual dysfunction. Four participants (one not using OCs) reported amenorrhea, and one participant (taking OCs) stated that menstrual changes were noticed relative to training load (bleed fewer days, menstruation ceasing, etc.).

Progesterone was low according to the reference range for the premenopausal luteal phase (<10.5–71.6 nmol·L^−1^) in 67% of the participants, with no trends between those considered “at risk” and “not at risk” for LEA according to LEAF-Q. Free T3 and estradiol were within the normal range for all participants. IGF-1 was elevated (>13.1–39.2 nmol·L^−1^) in 22% of female athletes, with those identified as “not at risk” according to LEAF-Q being within normal limits. However, menstrual cycle phase was not accounted for in this study and, therefore, these participants may have been within normal limits depending on the specific phase each athlete was in at the time of blood collection.

All male participants exhibited low TES, defined as 9–16.5 nmol·L^−1^ [5]. Six participants (67%) presented with clinically low TES (<9 nmol·L^−1^). IGF-1 was elevated in four participants (50%). There was no observed trend between low TES and IGF-1 levels. All participants were in the reference range for fT_3_. Estradiol levels were within the reference range of 12–56 pg·mL^−1^ [27].

Spinal DXA scans were not usable as most subjects had metal rods in this region, making it difficult for the software to distinguish between bone and metal, thus skewing the results for whole body scans. Therefore, hip z-scores were reported for all participants. Five female participants (56%) had clinically low BMD in the hip regional score (*Z* < −2), (one of which reported a bone-related injury within the past year), while two male participants (25%) had clinically low scores (*Z* < −2).

## 4. Discussion

This is one of the first studies to examine symptoms of LEA and RED-S related symptoms amongst national and international level para-athletes. Based on highly variable EA calculations and EDE-Q, the risk of RED-S appears to be low. Low BMD is more likely due to the SCI in this population, and the use of hormonal contraceptives was high, rendering BMD and reproductive hormone status inappropriate tools to use to assess chronic EA status. Parameters within the LEAF-Q may also be related to the SCI rather than chronic EA status, thus it appears to be disconnected from actual acute EA quantification in this population. Qualitative and quantitative measures showed considerable discrepancies that must be considered when interpreting the results and clearly demonstrate the need for the development of para-specific RED-S assessment tools. Finally, it should be noted that this was an exploratory investigation, and thus likely did not identify the true prevalence of LEA, but started to create controlled normative data values in these unique, understudied athletes.

### 4.1. Measured Energy Availability in Para-Athletes

Assessing EA in the field in AB athletes presents many opportunities for error, as calculating EA via EI and EEE recording is challenging and lacks sensitivity as a preferred diagnostic tool for the assessment of LEA [5,14]. While food records have been found to be the most preferred method of obtaining estimates of EI, the approach is also highly variable and prone to under-reporting, which may account for 10–45% variability in energy intake [30,31], especially when athletes are unfamiliar with the practice of intricate daily food recording of metrics around training [32]. It should be noted that some of the diet records (*n* = 5) were collected during a training camp in which athletes were provided meals by hotel catering and, thus, may not be reflective of their typical self-prepared diet. Conger and Bassett (2011) provided the only known compendium of energy costs of activity for individuals using wheelchairs [15]. Therefore, although the data may be outdated for current-day para-athletes, and are limited in the range of activities, the exercise mode in the compendium that most closely resembled that recorded during training was used to estimate exercise energy expenditure for each participant. Two female participants were considered to have moderate 7-day average EA (30–45 kcal·kg FFM^−1^·day^−1^), while the other six participants were considered to have optimal 7-day average EA (>45 kcal·kg FFM^−1^·day^−1^), but very large day to day variability existed, with some individuals demonstrating low EA on certain days (Figure 1). However, it should be noted that these thresholds for EA were developed in recreationally active AB females with full body functioning FFM, which may not be relevant for the para-athlete [14]. Indeed, most wheelchair athletes will present with lower body muscle atrophy, resulting in reduced whole-body FFM—thus, our calculated EA values in this cohort are likely to be artificially elevated in relation to AB clinical EA cut-offs. In comparison, Heikura et al. (2018) found that 31% of AB female distance runners had LEA and 69% of females had moderate EA, while no females had optimal EA. The authors also suggested that calculated EA was poorly correlated with other more chronic factors known to be associated with LEA, including reproductive, metabolic, and bone health [5]. No significant correlations were observed among EA and BMD in the current study. Out of the four male athletes who provided sufficient data to calculate EA, each athlete exhibited LEA (≤25 kcal·kg FFM^−1^·day^−1^) on one day out of the seven recorded days. These athletes displayed wide variability in EI and EEE across the week, so despite these single occurrences of LEA, the average EA across the week was well above the AB LEA threshold for each subject. Conversely, Egger et al. (2020) reported that the mean EA (25.1 ± 7.1 kcal·kg FFM^−1^·day^−1^) for a group of female wheelchair athletes (*n* = 6) was below the LEA threshold, while the males (*n* = 8) (36.1 ± 6.7 kcal·kg FFM^−1^·day^−1^) were above the threshold using AB cut-offs [14]. Research has suggested that short-term perturbations in EA do not appear to have as severe an impact on male athletes compared to female athletes [7,33]. However, recent research has suggested that extremely low EA, or even within day periods of LEA, can still result in measurable disturbances in hormone levels in male athletes after only eight days of intermittent LEA [34].

### 4.2. BMD Indicators of RED-S

Fifty-six percent of female participants, and twenty-five percent of the male participants presented with a hip *Z*-score < −2, which is clinically low, indicating a high risk for fracture [24,28]. Decreased BMD in the lower limbs is frequently a consequence of long-term wheelchair usage. Bones adapt to the prevailing conditions, i.e., a decrease in skeletal loading leads to a decrease in bone strength and density [11,35]. While it has been suggested that participation in sport appears to attenuate the expected loss of whole-body BMD by increasing the stress imposed on the skeleton at specific sites such as the forearm [36,37], osteoporosis is still present in nearly every individual with SCI. The ISCD recommends a DXA scan of the total hip, promixal tibia, and distal femur to diagnose osteoporosis every 2 years in individuals with SCI [24]. However, BMD of the distal radius may be a better indicator of LEA due to the decreased loading of the lower limbs in athletes with SCI [11]. Therefore, when assessing BMD values in relation to LEA in the para-athlete population, low BMD is more likely a factor related to the impairment, regardless of diet quality or energy intake. Therefore, BMD may be an inappropriate diagnostic criterion for assessing the risk of LEA in para-athletes.

### 4.3. Reproductive/Metbaolic Function in Para-Athletes

Menstrual dysfunction is an established indicator of LEA and RED-S [2]. Menstrual function in the female group was abnormal in four of the participants (44%), three of whom reported oral-contraceptive (OC) use. Furthermore, the majority (67%) of participants in this study were using some form of hormonal contraceptives being intrauterine or oral forms. Unless athletes are using a copper IUD, an assessment of true menstrual function linked to LEA is difficult in athletes taking OCs due to the impact of exogenous hormones. Research in 430 elite AB athletes indicated that 49.5% were using hormonal contraceptives and 69.8% had used them at some point. Proposed reasons for this use was related to difficulty in having a menstrual cycle during certain training and competition periods, along with the associated side-effects that exist with menstruation [38]. These menstrual concerns in AB athletes may be further amplified in a para-athlete population due to the added difficulties of mobility. The current study suggested that progesterone was low in 67% of female participants, while estradiol was within normal limits for each participant. The relationship between EA status and disruptions to endocrine function in female athletes is subject to within- and between-participant variability; however, the high prevalence of contraceptive use may explain the findings in the current study [2]. Therefore, menstruation patterns should be examined carefully, as menstruation and hormonal parameters are masked by contraceptive use and are likely not appropriate for assessing LEA in this population.

It has been suggested that 45–60% of males with chronic SCI present with low TES [39], and although the exact mechanism is unclear, it is suspected that disruption of the hypothalamic–pituitary–gonadal axis (HPG axis) related to spinal cord injury is partly responsible [39,40]. Sullivan et al. (2017) compared TES levels in 58 healthy males with complete, chronic (≥1 year) SCI against TES levels in a cohort of age-matched, able-bodied males; the researchers found that individuals with SCI are four times more likely to experience low TES [40]. In the present study, 100% of athletes with SCI (*n* = 5) exhibited low TES. Similar to the reported impaired bone health, it is plausible that the low TES was more closely related to the SCI rather than to LEA. To our knowledge, this is the first study to describe TES levels in male athletes with SCI.

None of the athletes in the present study exhibited low IGF-1. In fact, 50% of the participants had elevated IGF-1. Several studies examining IGF-1 and EA in AB male wrestlers and endurance athletes have demonstrated a suppression of IGF-1 when the athletes were in an energy deficit [4,6]; however, the picture is less clear when looking at individuals with SCI. It is common to observe depressed levels of IGF-1, independent from LEA, in individuals with SCI [41], but some studies have reported normal IGF-1 levels in this population [41,42]. It is possible that the athletes’ high activity levels help to maintain higher levels of IGF-1; however, additional research is needed to further support this hypothesis.

Previous research examining fT_3_ levels in AB athletes has reported low levels in athletes that also had LEA-associated low TES [15]; however, studies examining fT_3_ in SCI populations specifically reported normal levels [41,43]. The results of the present study are consistent with the latter findings, despite participants displaying low TES suggesting a possible variability in the presentation of risk factors of RED-S between able-bodied and para-athletes. It is important to note that the various clinical ranges discussed above were developed using able-bodied individuals, and therefore it may be inappropriate to apply the ranges to the SCI population.

### 4.4. Questionnaire Based Assessment of RED-S

According to the LEAF-Q, 78% of female participants were “at risk” for LEA based on a score ≥ 8, which may be skewed due to the responses received in the menstrual function section of the LEAF-Q. While no other known studies have used the LEAF-Q in para-athletes, Heikura et al. (2018) found that LEAF-Q scores differed in eumenorrheic (8.3 ± 3.7) and amenorrheic (12.8 ± 4.8) AB athletes. This significantly higher LEAF-Q score in the amenorrheic group led authors to conclude that LEAF-Q was an appropriate tool for assessing risk of the triad [5]. However, no trends existed between estimated energy availability and LEAF-Q scores in the current study.

In contrast to the LEAF-Q, the EDE-Q results suggest that one female participant and no male participants were at risk for disordered eating and potential low EA in this study. The reported low scores are consistent with previous research using the EDE-Q with male athlete populations. Torstveit et al. (2019) found only one out of 34 elite male endurance athletes to be at risk for disordered eating using the EDE-Q [43]. While it does not directly determine risk of low energy availability, this questionnaire has been considered an instrument of choice when identifying behaviors surrounding eating disorders. Out of the four subscales within the EDE-Q, participants scored highest in the “shape concern” category. In the para-athlete population specifically, this higher score regarding shape could have also been attributed to concern of fitting into their sport chair during competition. When asked (in separate questions) if participants restricted caloric intake due to concern of fitting into their sport chair or due to discomfort that may be felt when eating before activity, four female participants and one male participant reported restricting due to discomfort before activity, while three female participants reported restricting due to concern of fitting into their sport chair. It is plausible that concerns around the shape and size in SCI athletes may be related to the nature of their sport rather than to ED risk.

### 4.5. Limitations

This study has several limitations. The sample size is small and therefore larger samples are needed to confirm or challenge these results. EA data were incomplete for five of the male athletes due to incomplete diet and training logs. The EEE for several of the female participants was low, which may be due to a lighter phase of training. Additionally, all of the data were collected during a training camp for five of the athletes, which may have caused the athletes to deviate from their normal routines at times, possibly impacting both their EI and EEE. Furthermore, menstrual cycle phase was unaccounted for in this study, and may explain the female participants’ hormonal levels. Finally, the LEAF-Q has not been validated in this population of athletes and should be used with caution.

## 5. Conclusions

Considerable discrepancies existed between the results from the questionnaires and EA calculations in assessing risk of LEA in para-athletes. Additionally, quantitative screening tools (BMD and blood spot tests) may be difficult to use as diagnostic measures when assessing LEA until para-specific RED-S norms are developed. Studies that use DXA to examine bone characteristics should consider the sources of error and profound impact of the SCI that may obscure the integrity of the BMD measurements. Furthermore, the development of upper limb thresholds for BMD, or to examine different from expected BMD in a site that is well characterized (i.e., lumbar spine) may allow for a better observation of EA-induced BMD. Furthermore, the low BMD is more likely due to the SCI in this population, and the use of hormonal contraceptives was high, suggesting that BMD and reproductive hormone status may be inappropriate tools to use to assess chronic EA status in this population. This study concluded that when calculating EA based on dietary intake and EEE, no LEA existed and the risk for RED-S appears to be low in this para-athlete population. With very limited studies assessing EA in para-athletes, there is a lack of assessment tools specific to para-athletes that isolate symptoms associated with LEA compared to those associated with their physical impairment. This exploratory investigation has been one of the first to provide critically needed data in para-athletes in the pursuit of further developing validated norms in the assessment of RED-S in para-athlete populations.

The International Olympic Committee (IOC) has recognized the impact of energy status on physiological processes and supports the hypothesis that energy deficiency may contribute to menstrual dysfunction, low testosterone, impaired bone health, reproductive and hormonal imbalance, and ultimately impaired health and performance [44]. With differing energy requirements, bone health, and menstrual function, the ability to identify LEA in para-athletes will require population- and disability-specific assessment methodologies. Nevertheless, despite considerable variability, several LEA/RED-S indicators were found in some of our para-athlete cohort, and we would encourage the International Paralympic Committee (IPC) to further recognize the unique health and performance needs of para-athletes.

## Figures and Tables

**Figure 1 nutrients-13-00979-f001:**
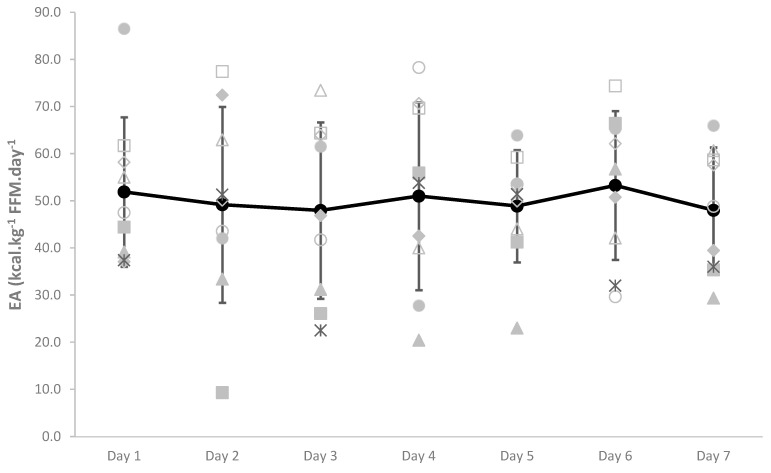
Within and between female participant daily variation in EA (kcal·kg^−1^ FFM·day^−1^). Daily mean ± SD EA for all participant is denoted in bold. Light gray dots represent individual participant data.

**Table 1 nutrients-13-00979-t001:** Method for assessing energy availability (EA) based on food/activity logs as follows EA = (energy intake (EI)—energy expended from exercise (EEE))/fat free mass (FFM).

Energy Intake (EI)	Seven-day consecutive food log completed by all participants to reflect dietary intake most representative of typical diet. Household weights, scales, and measures were used to record accurate portions sizes of meals (instructions included within food/activity log). A training video educating participants how to properly complete food logs (e.g., serving sizes, timing, food description) and importance of being precise was also implemented. RDN estimated total EI by analyzing food logs with dietary analysis software (ESHA).
Exercise Energy Expenditure (EEE)	Estimate EEE using a 7-day training log where exercise description, training duration, and intensity was recorded. Athletes encouraged to maintain normal routine during this time. Each exercise endeavor and training were assigned an energy cost (kcal·kg^−1^·hr^−1^) using a compendium of activities performed by wheelchair users, that represents the intensity and type of that activity [15]. Multiplied the energy cost for each training session by the duration of the session to yield EEE. REE was found using the Cunningham prediction equation and divided by 24 to get hourly REE [16]. Subtracted REE from tEEE so that only the additional energy cost of exercise was included in the EEE Use this EEE value in the equation above
Energy Availability (EA) cutoff values	Low EA: <30 kcal·kg FFM^−1^·day^−1^ Moderate EA: 30–45 kcal·kg FFM^−1^·day^−1^ Optimal EA: >45 kcal·kg FFM^−1^·day^−1^ [8]
Fat-free mass (FFM)	Fat-free mass was obtained from dual-energy X-ray absorptiometry (DXA) scans.

*Note*: METs = metabolic equivalents, tEEE = total EEE, ESHA = Elizabeth Stewart Hands and Associates. Table adapted from Heikura et al. [5].

**Table 2 nutrients-13-00979-t002:** Participant (*n* = 18) descriptive characteristics and dietary and training data.

**Females (*n* = 9)**																
	**1**	**2**		**3**		**4**	**5**			**6**		**7**	**8**	**9**		**Mean ± SD**
Age (yrs)	27	29		21		32	24			19		24	25	41		**27 ± 7**
Height (cm)	163	130		145		150	163			140		137	163	178		**152 ± 15**
Weight (kg)	44.0	36.8		42.0		42.3	54.5			55.1		34.1	57.0	64.5		**47.8 ± 10.3**
Duration of injury (yrs)	22	29		18		32	19			19		24	15	7		**21 ± 7**
Injury level/impairment	T-12	T-4		T-10		L1-2	L2-3			L3-4		L5	T-11	DAmp		**n/a**
Body Fat (%)	29.0	20.3		31.6		34.3	39.7			34.5		33.6	37.3	28.2		**34.0 ± 5.7**
EI (kcal·day^−1^)	1661	2026		1807		1679	1286			1975		1263	1941	2168		**1717 ± 280**
CHO (g·kg^−1^·day^−1^)	4.6	4.5		2.8		4.4	2.3			3.6		4.2	3.9	3.9		**3.7 ± 0.8**
PRO (g·kg^−1^·day^−1^)	1.9	3.7		2.7		1.7	1.3			1.6		1.3	1.9	1.6		**1.9 ± 0.9**
Fat (%kcal·day^−1^)	34	43		47		36	39			41		34	29	33		**38 ± 1**
Fiber (g·day^−1^)	30	24		9		17	21			15		10	21	22		**18 ± 7**
EEE (kcal·day^−1^)	110	78		113		41	191			580		40	233	549		**216 ± 196**
EA (kcal·kg^−1^ FFM·day^−1^)	49	67		59		59	**33**			**40**		54	49	**41**		**50 ± 11**
**Males (*n* = 9)**																
	**1**		**2**	**3**	**4**			**5**	**6**		**7**		**8**	**9**	**Mean ± SD**	
Age (yrs)	23		29	35	23			20	25		32		25	45	**27 ± 8**	
Height (cm)	168		168	183	180			175	163		142		175	160	**166 ± 5**	
Weight (kg)	60.3		51.9	74.2	63.2			64.8	71.3		75.0		52.0	68.2	**64.5 ± 8.7**	
Duration of Injury (yrs)	23		29	13	4			20	25		32		5	41	**21 ± 12**	
Injury level/impairment	SB		T10	T11	T4/T6			L3/L4	SB		CP		C6/C7	Poliy	**n/a**	
Body fat (%)	34.5		27.3	26.3	16.5			25.1	21.5		26.6		18.9	31.6	**25.4 ± 5.7**	
EI (kcal·day^−1^)	2459		2257	n/a	n/a			3695	2033		2390		n/a	n/a	**2566 ± 651**	
CHO (g·kg^−1^·day^−1^)	6.0		4.6	n/a	n/a			3.5	3.1		3.1		n/a	n/a	**4.1 ± 1.3**	
PRO (g·kg^−1^·day^−1^)	1.5		1.9	n/a	n/a			4.2	1.6		1.9		n/a	n/a	**2.2 ± 1.1**	
Fat (%kcal·day^−1^)	24		38	n/a	n/a			46	34		21		n/a	n/a	**33 ± 10**	
Fiber (g·day^−1^)	19		34	n/a	n/a			14	20		10		n/a	n/a	**22 ± 7**	
EEE (kcal·day^−1^)	156		n/a	n/a	190			n/a	52		n/a		n/a	249	**198 ± 47**	
EA (kcal·kg^−1^ FFM.day^−1^)	58		n/a	n/a	27			n/a	35		n/a		n/a	43	**41 ± 12**	

*Note.* Values are presented as means ± SD. CHO = carbohydrate; PRO = protein; LEAF-Q = Low Energy Availability in Females Questionnaire (35); EEE = exercise energy expenditure; EA = energy availability; DAmp = double amputee; SB = spina bifida; CP = cerebral palsy; Poliy = poliyomielyte; bold data indicate “at risk” or moderate EA in AB female subjects [8] and potentially at risk or moderate EA in AB male subjects based on emerging information [10].

**Table 3 nutrients-13-00979-t003:** Metabolic and reproductive hormone concentrations, bone density, and qualitative survey data for each participant.

**Females**										
	**1**	**2**	**3**	**4**	**5**	**6**	**7**	**8**	**9**	**Mean ± SD**
**Hormones**										
Estradiol (pg.mL^−1^)	55	12	54	49	21	35	56	13	101	44 ± 28
Progesterone(nmol.mL^−1^)	0.7	0.7	0.8	0.6	0.5	0.6	11.7	7.1	15.6	3.1 ± 4.1
IGF-1 (nmol.L^−1^)	35.3	34.3	20.3	27.8	53.7	31.2	25.6	43.2	17.3	32.1 ± 11.3
fT_3_ (pg.mL^−1^)	2.5	2.5	3.4	2.6	2.7	3.3	2.6	3.2	3.2	2.9 ± 0.4
**Bone Characteristics**										
Whole body (g.cm^2−1^)	0.97	0.99	1.51	1.32	1.39	0.96	0.88	0.93	1.06	1.11 ± 0.22
Hip z-score	**−2.2**	**−2.7**	**−1.0**	−0.1	**−2.1**	−0.9	**−3.3**	**−2.4**	−0.6	**−1.6 ± 1.2**
**Qualitative Surveys**										
LEAF-Q score	3	**15**	**12**	**9**	**8**	2	**9**	**10**	**12**	9 ± 4
EDE-Q Global score	0.72	0.24	0.09	0.08	**4.15**	0.48	0.09	2.3	0.27	0.93 ± 1.30
**Males**										
	**1**	**2**	**3**	**4**	**5**	**6**	**7**	**8**	**9**	**Mean ± SD**
**Hormones**										
Estradiol (pg.mL^−1^)	n/a	43	22	<10	20	36	45	41	38	32 ± 13
IGF-1 (nmol.L^−1^)	41.9	n/a	75	29.6	50.8	17.6	42.8	29.4	13.8	37.6 ± 19.7
fT_3_ (pg.mL^−1^)	3.1	3.2	3	3.6	3.2	3	2.5	3	3.4	3.1 ± 0.3
TES (nmol.L^−1^)	**8.5**	**10**	**11**	**4.3**	**7.6**	**7.8**	**5.7**	**10.2**	**5.9**	7.9 ± 2.3
**Bone Characteristics**										
Whole Body (g.cm^2−1^)	1.15	1.45	1.09	1.3	0.97	1.44	1.18	1.2	1.16	1.22 ± 0.16
Hip *Z*-Score	−0.9	**−2.5**	**−1.5**	**−1.3**	n/a	**−1.6**	**−1.3**	**−1.3**	**−2.9**	**−1.7 + 0.7**
**Qualitative Surveys**										
EDE-Q Global score	0.43	0.69	n/a	0	0	0.05	0.55	1.17	0.14	0.38 ± 0.41

Abbreviations: IGF-1 = insulin- like growth factor; fT_3_ = triiodothyronine; TES = testosterone; n/a = not available or missing data; BMD = bone mineral density; z-score = age-matched reference value for BMD, reference values: Z < −2, clinically low; Z ≤ −1, trend for low; Z > 1, normal [29]; EDE-Q = Eating Disorder Examination Questionnaire [18]. Bold data indicate risk factors associated with RED-S in AB subjects [2].

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
