# Peer review of "Risk of Low Energy Availability in National and International Level Paralympic Athletes: An Exploratory Investigation"

_nutrients, 2021, doi:10.3390/nu13030979_

Round 1

Reviewer 1 Report

Thank you for the opportunity to review this manuscript. The authors have attempted to explore the prevalence and risk of low energy availability in Paralympic athletes. Though the sample size is low, I believe this paper is a valuable contribution to the area and believe it may encourage further work in this area.  Overall, I think this is a well written paper but I'd like a couple of minor points to be addressed. Comments are below.

Introduction

A well written introduction which sets the scene appropriately.

Methods

Line 105 – Suggestion – for readers may be beneficial to say what the LEAF-Q is out of. I understand why you haven’t included it, but I think it may be beneficial.

Line 108 – Suggestion – same point for EDE-Q

Line 119 – What DXA analysis software was used?

Line 119 – Was the DXA scan analysis an issue for any of the athletes due to their injuries?

Line 126 – Hydration and glycogen variability can also be affected by the previous days activity. Were scans done following a training day? I understand for practicality why it may be mixed across the participants. 

Line 128 – Is this for both sexes?

Line 134 – First mention of measuring progesterone, should this be included on line 130 after estradiol? Furthermore, if not controlling for phase of menstrual cycle is there any point in this measure? I accept I have a lack of expertise in this specific area. 

Results

Line 199 – This relates to a previous point, perhaps worth mentioning in the methods.

Line 206 – Relates to a previous point. Please be clear in your methods which hormones you are measuring for each sex and why.

Discussion

Line 214 – I think this is a really good opening paragraph which summaries the outcomes of this study well.

Line 268 – Typically I don’t believe a sentence should start with a number as a figure. Should be written out as Fifty-sex percent of… However I understand that this may not in the journal requirements. Just a suggestion.

Line 276 – Typo. ‘every to 2 years’.

Line 296 – Progesterone was only measured in females.

Line 329 – No mention of cortisol in discussion. Is it needed in results?

Line 340 – Needs to be separate from section 4.4. Perhaps have a limitations section.

Author Response

Reviewer #1

Thank you for the opportunity to review this manuscript. The authors have attempted to explore the prevalence and risk of low energy availability in Paralympic athletes. Though the sample size is low, I believe this paper is a valuable contribution to the area and believe it may encourage further work in this area.  Overall, I think this is a well written paper but I'd like a couple of minor points to be addressed. Comments are below.

Author comments: Thank you for the suggestions. We believe the paper has improved due to the reviewer input.

Introduction

A well written introduction which sets the scene appropriately.

Methods

Line 105 – Suggestion – for readers may be beneficial to say what the LEAF-Q is out of. I understand why you haven’t included it, but I think it may be beneficial.

Thank you for the suggestion. We have included that this is a 25 item questionnaire in the paper. However, other papers do not report the upper score limits on these surveys.

Line 108 – Suggestion – same point for EDE-Q

Thank you for the suggestion. We have included this info. However, other papers do not report the upper score limits on these surveys.

Line 119 – What DXA analysis software was used?

This is included on line 120 – the General Electric, Lunar iDXA in Daytona & Hologic QDR 4500A in Urbana

Line 119 – Was the DXA scan analysis an issue for any of the athletes due to their injuries?

That’s a great question. Most of these athletes have a routine DXA, so they were familiar with the process. Line 200 expresses the major situation in terms of difficulty due to spinal rods. The uniqueness of the athletes morphology, ie., scoliosis and hip flexion, are a consideration for performing DXA in this population but did not impede the hip scan.

Line 126 – Hydration and glycogen variability can also be affected by the previous days activity. Were scans done following a training day? I understand for practicality why it may be mixed across the participants. 

Great point. All scans were performed after a training day. All scans were performed fasted and before training, no participants were scanned following a rest day. 

Line 128 – Is this for both sexes?

Yes, that’s correct. We included a statement to reflect this.

Line 134 – First mention of measuring progesterone, should this be included on line 130 after estradiol? Furthermore, if not controlling for phase of menstrual cycle is there any point in this measure? I accept I have a lack of expertise in this specific area. 

Thank you. We have included this. This measure still provides a snapshot as to whether estradiol is within the normal range or not.

Results

Line 199 – This relates to a previous point, perhaps worth mentioning in the methods.

Thank you. We have included this. We also mention in our intro that hormonal levels of estrogen, testosterone and IGF may be decreased as a result of LEA.

Line 206 – Relates to a previous point. Please be clear in your methods which hormones you are measuring for each sex and why.

Thank you. We have clarified this.

Discussion

Line 214 – I think this is a really good opening paragraph which summaries the outcomes of this study well.

Thank you.

Line 268 – Typically I don’t believe a sentence should start with a number as a figure. Should be written out as Fifty-sex percent of… However I understand that this may not in the journal requirements. Just a suggestion.

Thank you. We have written the percentage out.

Line 276 – Typo. ‘every to 2 years’.

Thank you.

Line 296 – Progesterone was only measured in females.

We actually measure in all participants but only reported in the females.

Line 329 – No mention of cortisol in discussion. Is it needed in results?

Good point. We have deleted cortisol.

Line 340 – Needs to be separate from section 4.4. Perhaps have a limitations section.

Thank you. We moved this to the limitations section for clarity

Reviewer 2 Report

This was a well-written article that adds valuable descriptive data, which is currently lacking in this population.  There were several limitations to the design but all were acknowledged and the data were interpreted accordingly.  My suggestions are minor. 

In line 41, "para athlete" should be hyphenated to match the other occurrences of this term throughout the paper.  I also suggest adding the definition of what constitutes this population at this point in the paper. 

In the inclusion criteria in lines 79-80, this is described but it's not clear if that is the definition for what constitutes a para-athlete "universally" 

On lines 145 and 146, the term "para athletics" was used to describe 7 of the females and 5 males.  Can you describe what activities that involves?  I suspect some readers won't know.   

In Table 2, I really appreciated the data you presented.  The "duration of injury" rows were interesting.  I recommend adding some points about this in the discussion, especially for the two males and one female who were only injured in the last 4, 5, and 7 years which is much different than the other participants. 

Please add the point from lines 188-189 that menstrual cycle was unaccounted for in this study to the limitations section starting on line 340. 

Finally, do you have distal radius BMD scores available from your DEXA scans?   If so I suggest adding them given the points you raised about hip BMD not being the best measure in this population as it's not a weight-bearing region.

Author Response

Reviewer #2

This was a well-written article that adds valuable descriptive data, which is currently lacking in this population.  There were several limitations to the design but all were acknowledged and the data were interpreted accordingly.  My suggestions are minor. 

Author comments: Thank you for the suggestions. We believe the paper has improved due to the reviewer input.

In line 41, "para athlete" should be hyphenated to match the other occurrences of this term throughout the paper.  I also suggest adding the definition of what constitutes this population at this point in the paper. 

In the inclusion criteria in lines 79-80, this is described but it's not clear if that is the definition for what constitutes a para-athlete "universally" 

We have included the following sentence: “Para athletes are defined by the International Paralympic Committee (IPC) to be an athlete with visual, physical, or intellectual impairment.” Therefore, this paper will refer to competitive athletes competing in a specific IPC classification category as para athletes.

On lines 145 and 146, the term "para athletics" was used to describe 7 of the females and 5 males.  Can you describe what activities that involves?  I suspect some readers won't know.   

Thank you. We provide a statement for clarity.

In Table 2, I really appreciated the data you presented.  The "duration of injury" rows were interesting.  I recommend adding some points about this in the discussion, especially for the two males and one female who were only injured in the last 4, 5, and 7 years which is much different than the other participants.

Thank you for the suggestion. However, when looking at the individual data and duration of injury, we did not notice any trends that are worth noting.

Please add the point from lines 188-189 that menstrual cycle was unaccounted for in this study to the limitations section starting on line 340. 

Thank you. We have included this information in the limitations section as well

Finally, do you have distal radius BMD scores available from your DEXA scans?   If so I suggest adding them given the points you raised about hip BMD not being the best measure in this population as it's not a weight-bearing region.

In retrospect that would have been ideal. However, we have not comparative data on this.